# Learning Dictionary for Visual Attention

**Yingjie Liu**[1]    **Xuan Liu**[2]    **Hui Yu**[2]    **Xuan Tang**[1]    **Xian Wei**[1] *

[1]East China Normal University
[2]FJIRSM, Chinese Academy of Sciences

## Abstract

Recently, the attention mechanism has shown outstanding competence in capturing global structure information and long-range relationships within data, thus enhancing the performance of deep vision models on various computer vision tasks. In this work, we propose a novel alternative dictionary learning-based attention (*Dic-Attn*) module, which models this issue as a decomposition and reconstruction problem with the sparsity prior, inspired by sparse coding in the human visual perception system. The proposed *Dic-Attn* module decomposes the input into a dictionary and corresponding sparse representations, allowing for the disentanglement of underlying nonlinear structural information in visual data and the reconstruction of an attention embedding. By applying transformation operations in the spatial and channel domains, the module dynamically selects the dictionary's atoms and sparse representations. Finally, the updated dictionary and sparse representations capture the global contextual information and reconstruct the attention maps. The proposed *Dic-Attn* module is designed with plug-and-play compatibility and facilitates integration into deep attention encoders. Our approach offers an intuitive and elegant means to exploit the discriminative information from data, promoting visual attention construction. Extensive experimental results on various computer vision tasks, e.g., image and point cloud classification, validate that our method achieves promising performance, and shows a strong competitive comparison with state-of-the-art attention methods.

## 1  Introduction

Visual perception plays a critical role in obtaining external information. However, processing massive visual information in the real world is challenging for both human brains and computers. Modern psychology and cognitive neuroscience find that the visual attention mechanism is one of the essential keys to a human's good cognitive ability [43], enabling selective and sparse activation of neurons in response to input. In the field of computer vision (CV), various attention methods have also been developed, with successful applications in lots of Convolutional Neural Networks and Transformers, revealing great potential in visual tasks [4, 32, 5, 48, 46, 15]. CV attention methods utilize various calculation functions to generate attention maps, but they can be seen as a general form (refer to Sec. 3.1) [11, 41, 10]. Researchers have recognized several advantages of the attention mechanism, e.g., capturing global long-term information and extracting hierarchical data structures [6]. Moreover, recent research has also highlighted the potential of leveraging sparsity to develop attention algorithms [31, 62, 23]. However, there are still pending questions regarding the efficiency and effectiveness of current attention methods when dealing with regular image features and non-Euclidean point cloud data [57, 35, 9]. Considering the significant memory costs and time complexity [42] associated with current attention methods, it is crucial to explore possibilities of optimizing their internal operations or re-evaluating the current form, especially in the view of sparsity.

---

*Corresponding author: `xwei@sei.ecnu.edu.cn`

In this context, recalling the previous works and considering the original motivation behind introducing attention methods remind us of the dictionary learning/sparse coding algorithm, which holds importance but is often overlooked in present discussions. The dictionary learning algorithm decomposes visual data into a dictionary basis and their corresponding sparse representations/codes. Under different conditions, the sparsity of sparse code can be adapted: code with higher sparsity (more zero elements) mostly corresponds to features of smaller scale, higher roughness, or higher hierarchy. Furthermore, this process echoes the nonlinear effects in complex cells and the sparse coding strategies for sensory information in the primary visual cortex and advanced visual region [34, 1, 19, 56]. It has been observed that the structure in a sparse domain could make the hidden patterns more prominent and easier to be captured [29, 52, 55, 60]. All these motivate us to develop an alternative novel attention form, i.e. **Dictionary learning-based Attention** (*Dic-Attn*), that offers a visual focus for deep neural networks (DNNs). The *Dic-Attn* module can be conveniently used to replace existing attention modules, and compatibly explain their working principles.

The *Dic-Attn* module employs two selection matrices to exploit channel and spatial attention from dictionary atoms. By selecting the appropriate basis and re-locating the coordinates, we can uncover hidden visual attention information by reconstructing the attention map using the updated basis (dictionary) and coordinates (corresponding sparse representations). Note that, the sparse representations in our proposed module originate from the non-linear decomposition of visual data and are subsequently reconstructed into the attention matrix. The sparsity property of the attention matrix is induced naturally by the sparse prior [52, 50, 60] and is related to the visual attention area within the images. The main contributions of this paper are summarized as follows:

1. We propose a novel visual attention module, namely, the **Dictionary learning-based Attention** (*Dic-Attn*) module, which is based on dictionary learning and provides better interpretability of providing contextual information.

2. We demonstrate that the proposed module has great potential in disentangling non-linear structural information by employing a masked dictionary and transformed sparse representations. We construct a deep attention stage composed of the proposed modules. Each module at different depths, ranging from shallow to deep, dynamically updates dictionaries and sparse representations through spatial and channel transformations, step-wise-ly reconstructing the final attention map.

3. We apply the proposed module to various visual models since it is designed to be plug-and-play, achieving better performance than various existing attention modules on image classification, semantic segmentation, and point cloud classification, with fewer parameters and faster computation.

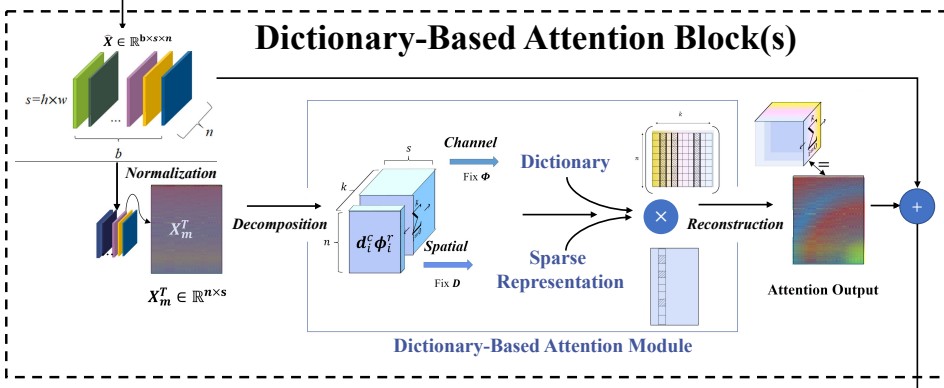

Figure 1: The pipeline of our proposed *Dic-Attn* block. First, input features $\hat{\mathbf{X}}$ into the *Dic-Attn* block. Within the *Dic-Attn* block, the inner *Dic-Attn* module decomposes the normalized input $\mathbf{X}$ into a dictionary and corresponding sparse representations, which will be transformed in the spatial and channel domains. The *Dic-Attn* module can dynamically adjust the weights to promote the visual focus on influential elements. This process is both data and task-driven, thereby enhancing the effectiveness of the attention mechanism. Finally, the output of the *Dic-Attn* block is obtained by summing the input with the **Attention Output**. The *Dic-Attn* block can be stacked layer by layer to form a deep attention model, enabling capturing and integrating visual attention hierarchically across multiple levels.

## 2 Related Work

**Attention Mechanism in Computer Vision Tasks.** CV Attention methods simulate the phenomenon of the human visual system, where selective attention helps neural system analyze and understand complex scenes more efficiently. Recurrent Model [32] is a pioneering work that develops the attention mechanism in DNNs. Later, various attention methods are proposed. Some of them try to improve the pre-processing method of the input images [8, 54], while others modify the way of obtaining the attention matrix, resulting in a series of variants, e.g., spatial transformer network (STN) [16], self-attention (SA) [46], squeeze-and-excitation network (SENet) [14], dual attention (DA) module [5], double attention ($A^2$) module [4], hamburger (HAM) module [6], etc. [21, 10, 48, 25, 39]. The SA module is proposed in [46] and first introduced to computer vision in [49]. It achieved great success in various vision tasks and made the Vision Transformer (ViT) successful in academia. [6] constructs the long-term global context learning as a low-rank recovery problem and proposes the matrix decomposition (MD) module, focusing only on the low-rank property prior assumption of feature representations. It may have limitations in fully capturing the complexity and richness of global structure information in the data. Most recent related work [39] was inspired by the two prominent features of the human visual attention mechanism, i.e., recurrency and sparsity. The so-called recurrency attribute in [39] naturally comes from the ordinary differential equation description of the sparse coding update process, which is actually inside the attention module and the sparse representations require a multi-step update. It also lacks the analysis of attention modules with different depths. Moreover, the residual connections between multi-layer attention modules still remain under explanation. [23] measures sparsity as the percentage of nonzero entries in the activation maps of each layer. It finds that sparsity emerges in the parameters of all layers of the Transformer. However, it does not explicitly state whether the output of the attention module exhibits sparsity or not. Some studies [37, 51, 17, 33] have tried to post-hoc-interpret these developed attention methods, and utilize white-box approaches to do the analysis [13, 41, 58, 31].

**Dictionary Learning in Computer Vision Tasks.** Sparsity is an important and noticeable prior in the field of computer vision, e.g., the statistically redundant property of natural images is served for dictionary learning. Hence, image sample $\mathbf{x}$ is reconstructed with the dictionary $\mathbf{D}$ and its corresponding sparse code $\phi$, i.e., samples can be decomposed into a dictionary and corresponding sparse representations. The dictionary $\mathbf{D}$ captures representative and critical features from the inputs and reveals the interference of insignificant information. Dictionary learning has been extensively studied and verified to be an outstanding approach for exploiting the underlying sparse structures hidden in images [29, 52, 50, 55, 60], efficiently transferring and expressing information from high-dimensional image data into sparse space. It plays an important role in various computer vision tasks, e.g., image reconstruction [1, 29, 3, 38, 64], image classification[52, 47, 55, 24, 60], visual tracking [61] and others.[39]. Note that, compared with non-negative matrix factorization and vector quantization applied in MD module [6] mentioned above, the dictionary learning algorithm performs better in the case of natural images [26]. Technical details about how we learn the dictionary is shown in Sec. 3.2.

## 3 Method

In this section, we present the proposed dictionary learning-based visual attention (Dic-Attn) method. First, we give a brief review and summary of the CV attention form and dictionary learning. Then, we describe the main idea of the *Dic-Attn* module to obtain visual attention. Finally, we analyze the core concept behind the *Dic-Attn* module. The overall structure of the proposed *Dic-Attn* method is illustrated in Fig. 1.

### 3.1 Attention Form

Generally speaking, we can summarize a CV attention form [11, 41].

$$\textbf{Input} : \mathbf{X},$$
$$\textbf{Attention Module} : f(\mathbf{X}).$$
$$\textbf{Attention Block} : g(f(\mathbf{X}), \mathbf{X}).$$

The input features $\mathbf{X}$ are firstly pre-processed. Subsequently, the attention module obtains an attention map of the input. Finally, $g(\cdot)$ is introduced to further connect attention map and input features, e.g., the normalization operation, drop operation, skip-connection, etc.. All these constitute an **Attention Block**. Among them, the function $f(\mathbf{X}) : \mathbf{X} \mapsto$ **Attention Output** in the attention module is the most critical mapping. Attention blocks can be added to the middle of the network, or form a feed-forward network. For example, in the ViT encoder with depth $N$, there will be $N$ attention blocks. Take ViT as an example, suppose there is a feature tensor $\mathbf{Z}$ with shape $b \times n \times h \times w$, where $n$ denotes the number of channels. The feature tensor is vectorized in the spatial domain and the last two dimensions are transposed. Hence, the attention block takes a sequence of image features as input, i.e., $\mathbf{X} \in \mathbb{R}^{b \times s \times n}$, where $s = h \times w$. The Self-Attention module is introduced and subsequently outputs a spatial attention map, essentially operating in the spatial domain. The formulation of the standard Self-Attention module in ViT [46] is as follows:

$$
\begin{aligned}
[\mathbf{Q}, \mathbf{K}, \mathbf{V}] &= [\mathbf{W_Q X}, \mathbf{W_K X}, \mathbf{W_V X}], \\
\mathbf{A} &= softmax(\mathbf{W}_{scale}(\mathbf{Q K}^T)), \\
\textbf{Attention Output} &= \mathbf{AV},
\end{aligned}
\tag{1}
$$

where $\mathbf{X} \in \mathbb{R}^{b \times s \times n}$ is the input, $\mathbf{Q} \in \mathbb{R}^{b \times s \times n}$, $\mathbf{K} \in \mathbb{R}^{b \times s \times n}$ and $\mathbf{V} \in \mathbb{R}^{b \times s \times n}$ denote the query matrix, the key matrix, and the value matrix, respectively. They are obtained by full-connected layers. $\mathbf{W_Q}, \mathbf{W_K}$ and $\mathbf{W_V}$ are all square matrices with the shape $n \times n$. $\mathbf{A}$ represents the attention matrix. Finally, through the residual structure, i.e. skip-connection, attention blocks have the final output result, i.e., **Attention Output** $+ \mathbf{X}$.

The implicit variables or the spatial attention map results of these attention modules can be visualized, e.g., traditional SA blocks in ViT [46], and External-Attention (EA) blocks [10]. These works indicate that the attention module focuses on the discriminative regions of each image. Recalling the CV attention form we summarized at the beginning of this section, we can find that $f(\cdot)$ is the most important function when obtaining visual attention in DNNs. In this paper, $f(\cdot)$ contains the process of dictionary learning, the transformation of both learned dictionary & corresponding sparse representations, as well as reconstruction, which is to be introduced in Sec. 3.2.

### 3.2 Dictionary Learning

Formally, given data $\mathcal{X} := [x_1, ..., x_n] \in \mathbb{R}^{n \times s}$, the standard dictionary learning algorithm is to find a dictionary $\mathbf{D}$ and the corresponding sparse code $\mathbf{\Phi}$. The dictionary $\mathbf{D} = [\mathbf{d}_1, ..., \mathbf{d}_k]$ has a shape of $n \times k$, the sparse code $\mathbf{\Phi} = [\boldsymbol{\phi}_1, ..., \boldsymbol{\phi}_k] \in \mathbb{R}^{k \times s}$ selects atoms from the dictionary to reconstruct the input $\mathcal{X}$. The objective function of dictionary learning can be formulated as follows:

$$
\begin{aligned}
\underset{\mathbf{D}, \boldsymbol{\phi}_i}{\operatorname{argmin}} \ & h(\mathbf{D}) + \sum_{i=1}^{s} \frac{1}{2} \|x_i - \mathbf{D}\boldsymbol{\phi}_i\|_2^2 + g(\boldsymbol{\phi}_i), \\
& \text{s.t. } \mathbf{D} \in \mathcal{S}(n, k), \boldsymbol{\phi}_i \in \mathbb{R}^k,
\end{aligned}
\tag{2}
$$

where $h(\mathbf{D}) = \alpha \sum_{i \neq j} \left\| \mathbf{d}_i^T \mathbf{d}_j \right\|_F^2$ denotes the regularization term of the dictionary. It leads to mutual incoherence between pair atoms in the dictionary. Besides, $\mathbf{D} \in \mathcal{S}(n, k) := \left\{ \mathbf{D} \in \mathbb{R}_*^{n \times k} : diag(\mathbf{D}^T \mathbf{D}) = \mathbf{I}_k \right\}$. Namely, the set of dictionaries is a product manifold of $s$ times the $(k-1)$-dimensional unit sphere, i.e., $\mathcal{S}(n, k)$ restrict all atoms $\mathbf{d}_i \in \mathbb{R}^k$ to have unit norm. $g(\boldsymbol{\phi}_i)$ usually utilizes norm constraint to control the sparsity of the sparse code $\boldsymbol{\phi}_i$. The $l_1$ norm regularization is a popular way to replace $l_0$-norm by $l_1$-norm convex relaxation, known as the Lasso [44] regression. It has numerous variants, e.g., elastic net. Elastic net inherits the stability of Ridge and the sparsity brought by Lasso, for combining both regularization terms of $l_1$ norm and $l_2$ norm. Hence in this paper, we have $g(\boldsymbol{\phi}_i) = \beta \|\boldsymbol{\phi}_i\|_1 + \frac{\lambda}{2} \|\boldsymbol{\phi}_i\|_2^2$. $\beta$ and $\lambda$ are regularization weights.

Eq. (2) can be regarded as an iterative optimization problem, where dictionary $\mathbf{D}$ and sparse representation alternately updates. Each sparse code is an implicit function about the temporarily fixed dictionary $\mathbf{D}$. With Eq. (2), the incoherent assumption of dictionary atoms and the sparse prior $g(\boldsymbol{\phi}_i)$ [50], the close solution of sparse representation can be expressed as follows:

$$
\boldsymbol{\phi}_i^* = \left(\mathbf{D}_\Lambda^T \mathbf{D}_\Lambda + \lambda \mathbf{I}\right)^{-1} \left(\mathbf{D}_\Lambda^T x_i + \beta \mathbf{v}_\Lambda\right),
\tag{3}
$$

where $\Lambda := \{i \in \{1, ..., s\} | \phi_{ij}^* \neq 0\}$ denotes the set of indexes of the non-zero entries of the solution $\phi_i^* = [\phi_{i1}^*, .., \phi_{is}^*]$. $\mathbf{D}_\Lambda$ is the subset of the dictionary in which the indexes of atoms fall into $\Lambda$. $|\Lambda|$ denotes the sparsity of $\phi_i^*$, $\mathbf{v}_\Lambda \in \{\pm 1\}^{|\Lambda|}$ carries the signs of $\phi_i^*$. Technically, it is easy to realize the calculation of the above polynomial formula and obtain the sparse representations.

Constructing or learning a dictionary is also crucial for achieving the goal of highlighting significant objects while suppressing distracting or noisy elements. In the early stage, the dictionary is constructed by Cosine functions, Fourier, Wavelets, Contourlets, Gabor, a set of complete bases, etc. These approaches were widely employed in signal processing due to their mathematical simplicity. However, such a fixed dictionary is manually designed under some mathematical constraints and is not flexible enough to represent complex natural image structures. Recently, researchers have turned to directly initializing and learning the dictionary from image data, which can also improve the sparsity of $\mathbf{\Phi}$ and the performance of downstream tasks [52, 60, 64]. The dictionary contains prototype underlying features of vision data, based on which each sample can find its unique optimal sparse solution.

### 3.3 The Proposed *Dic-Attn* for Visual Attention

#### 3.3.1 *Dic-Attn* Method

The *Dic-Attn* module first decomposes the input into a dictionary and sparse representation. Then its primary objective is to provide visual attention to the model by reconstructing the attention map with residuals. Its process is as follows,

$$
\begin{aligned}
\mathbf{X} &= \mathbf{D\Phi}, \\
\mathbf{V} &= \mathbf{DW_D}, \\
\mathbf{A} &= softmax(\mathbf{W}_{scale}(\mathbf{W_\Phi \Phi})), \\
\textbf{Attention Output} &= \mathbf{VA},
\end{aligned}
\tag{4}
$$

where $\mathbf{X} = [\mathbf{X}_1^T, \mathbf{X}_2^T, ..., \mathbf{X}_b^T] \in \mathbb{R}^{b \times s \times n}$. Each atom $\mathbf{d}_i \in \mathbb{R}^n$ in the dictionary $\mathbf{D} \in \mathbb{R}^{n \times k}$ will be re-weighted by $\mathbf{W_D} \in \mathbb{R}^k$. The elements of $\mathbf{W_D} \in \mathbb{R}^k$ are trainable mask weights for dictionary atoms. $\mathbf{W_\Phi} \in \mathbb{R}^{k \times k}$ denotes the selection mask matrix for sparse representations $\mathbf{\Phi} = [\mathbf{\Phi}_1, ..., \mathbf{\Phi}_b] \in \mathbb{R}^{b \times k \times s}$. $\mathbf{W}_{scale}$ denotes the scaling matrix. Note that $k = \alpha n$, $\alpha$ is the hyper-parameter.

Firstly, the dictionary is learned on the entire training-dataset, aiming to be adaptive to all inputs $\mathbf{X}$. It contains diverse and important features (in atoms) of the input, so it can solve the sparse coding problem and obtain the corresponding code $\mathbf{\Phi}$. We have:

$$
\mathbf{D} = [\mathbf{d}_1^c \quad \mathbf{d}_2^c \quad \cdots \quad \mathbf{d}_k^c] = [\mathbf{d}_1^r \quad \mathbf{d}_2^r \quad \cdots \quad \mathbf{d}_n^r]^T,
$$

therefore,

$$
\mathbf{\Phi}_m = [\phi_1 \quad \phi_2 \quad \cdots \quad \phi_s] = [\psi_1 \quad \psi_2 \quad \cdots \quad \psi_k]^T,
$$

$$
\mathbf{X}_m =
\begin{bmatrix}
\mathbf{x}_{11} & \mathbf{x}_{12} & \cdots & \mathbf{x}_{1n} \\
\mathbf{x}_{21} & \mathbf{x}_{22} & \cdots & \mathbf{x}_{2n} \\
\vdots & \vdots & \vdots & \vdots \\
\mathbf{x}_{s1} & \mathbf{x}_{s2} & \cdots & \mathbf{x}_{sn}
\end{bmatrix}
=
\begin{bmatrix}
\mathbf{d}_1^r \phi_1 & \mathbf{d}_1^r \phi_2 & \cdots & \mathbf{d}_1^r \phi_n \\
\mathbf{d}_2^r \phi_1 & \mathbf{d}_2^r \phi_2 & \cdots & \mathbf{d}_2^r \phi_n \\
\vdots & \vdots & \vdots & \vdots \\
\mathbf{d}_s^r \phi_1 & \mathbf{d}_s^r \phi_2 & \cdots & \mathbf{d}_s^r \phi_{n},
\end{bmatrix}
\tag{5}
$$

$$
\mathbf{x}_{ij} = \mathbf{d}_i^r \phi_j = \sum_{t=0}^{k} (\mathbf{d}_t^c \psi_t)_{ij},
$$

where $\mathbf{X}_m$ denotes the $m$-th batch of $\mathbf{X}$, $\mathbf{\Phi}_m = [\phi_1, ..., \phi_n] = [\psi_1, ..., \psi_k]^T \in \mathbb{R}^{k \times s}$, $\mathbf{d}_t^c$ and $\mathbf{d}_i^r$ denotes the $t$-th column and the $i$-th row of $\mathbf{D}$. The optimization process of dictionary learning maximizes the descriptiveness of the atoms $\mathbf{d}_t^c$ while minimizing the redundancy between original inputs $\mathbf{X}$ and $\mathbf{D\Phi}$. Then, transformations $\mathbf{W_D}$ and $\mathbf{W_\Phi}$ for dictionaries and sparse representations are introduced, respectively, which facilitate the integration of spatial and channel attention into the model by assigning weights in both the spatial and channel domains.

The diagonal transformed matrix $\mathbf{W_D}$ and transformed matrix $\mathbf{W_\Phi}$ are specifically designed to re-weight the dictionary and transform its corresponding sparse representations, respectively. Through

**Algorithm 1:** Dictionary Learning-Based Attention Module

---

**Input**: $\mathbf{X} \in \mathbb{R}^{b \times s \times n}$, depth $N$,
**Parameter**: $\mathbf{D} \in \mathcal{S}(n,k), \mathbf{W_\Phi} \in \mathbb{R}^{k \times k}, \mathbf{W_D} \in \mathbb{R}^k, \mathbf{W}_{scale}$
**Latent Variables**: $\mathbf{\Phi} \in \mathbb{R}^{k \times s}, f(\mathbf{\Phi^T}), \mathbf{V}$
**Output**: **Attention Output**
\# Initialize $\mathbf{D} \in \mathcal{S}(n,k)$ by randomly sampling from $\mathcal{N}(0,1)$ and atom-wise normalization;
\# Initialize $\mathbf{W_\Phi}, \mathbf{W_D}, \mathbf{W}_{scale}$ with reference to kaiming-init method [12];
\# **forward**
\#\# Update $\mathbf{\Phi}$ by Eq. (3);
\#\# Obtain **Attention Output** by Eq. (4);
\# **backward**
\#\# Update $\mathbf{D}, \mathbf{W_\Phi}, \mathbf{W_D}, \mathbf{W}_{scale}$ by minimizing loss function of the network ;

---

the transformation of the dictionary, spatial attention is introduced during the reconstruction process, effectively leveraging the discriminative power of atoms [18, 2, 64, 27]. Different sizes of the diagonal elements in the $\mathbf{W_D}$ impose weights on different channels $\mathbf{d}_i \psi_i$ in the reconstruction process. In view of sparse representations/codes, the *Dic-Attn* module employs sparse prior and exploits the intrinsic underlying structure of features. The sparsity involved in the attention module arises spontaneously from the statistically redundant natural images. The sparse representation serves as the coordinates of the dictionary. These sparse representations indicate the importance or relevance of each atom in the dictionary for reconstructing the input features. Importantly, the attention similarity between different tokens or grid features in $\mathbf{X}$ is inherently encoded within the dictionary and the sparse representation. $\mathbf{W_\Phi}$ modifies individual elements and $\mathbf{W}_{scale}$ applies unified weights within each column of $\mathbf{\Phi}$, respectively. These transformation parameters are driven by the task objective function and are updated through the backpropagation process.

Algorithm 1 summarizes the algorithm flow of the *Dic-Attn* module. Furthermore, it is worth noting that constructing a deep *Dic-Attn* vision model is feasible. All these transformation matrices are trained in a back-forward manner and serve as a crucial link in connecting the dictionary, sparse representations, and reconstructed attention maps. To enhance the stability and effectiveness of the *Dic-Attn* module, it is recommended to include a normalization step before the attention module as [46]. Additionally, a residual connection between the input $\mathbf{X}$ and the **Attention Ouptut** facilitates the flow of information. This results in the formation of the *Dic-Attn* block, as illustrated in Fig. 1.

In summary, the *Dic-Attn* module utilizes a data-initialized and task-driven dictionary strategy, which captures global information and serves as a comprehensive resource containing essential data constituents. The process of reconstructing attention maps fully capitalizes on the dictionary learning sparse coding approach, the ability to extract both channel and spatial information from nonlinear features. The *Dic-Attn* module has a simpler structure with fewer parameters and elegant internal operations while achieving higher performance. Moreover, the *Dic-Attn* module is a convenient plug-and-play attention module that can be easily integrated into various vision networks.

### 3.3.2 General *Dic-Attn*-Based Vision Transformer

Herein, we give a guide on how to build a *Dic-Attn*-based Vision Transformer and analyze the role of the proposed *Dic-Attn* method. The model architecture can be divided into three main parts: the preprocessing stage, the deep attention stage, and the post-processing stage for downstream tasks.

Firstly, multi-layer convolution and linear operations are employed to extract primary visual features from the natural images. Then, multiple concatenated *Dic-Attn* blocks form the deep attention stage, generating the final attention map step by step. To conveniently introduce this process, we further summarize the *Dic-Attn* form as follows

$$Reconstruction(\widetilde{\mathbf{D}}, \mathbf{NL}(\mathbf{\Phi})), \tag{6}$$

where $\mathbf{NL}(\cdot)$ denotes non-linear scaling function. $\widetilde{\mathbf{D}}$ is the learned dictionary with re-weighted atoms. $\mathbf{NL}(\mathbf{\Phi})$ can be treated as a response prediction model [56]. We assume that the input of $(l+1)$-layer attention block is **Attention Output**$_l + \mathbf{X}_l$, i.e., the output of the previous layer's attention block. Learning-based dictionary $\mathbf{D}$ of each *Dic-Attn* module groups significant features of different objects

in the input, and the most salient objects are preserved while those irrelevant are suppressed by further selecting sparse representations. The output of each layer has a common intersection as the input $\mathbf{X}_1$ of the first layer, which is a task-independent bias for dictionary learning in *Dic-Attn* modules of any depth. Moreover, the output can be further considered as consisting of two components, one of which is the $l$-layer module's attention area, and the other component represents the ignored part. The first component will affect deep attention modules layer by layer, and adjust the task-driven selection matrix through backpropagation. Hence, our proposed *Dic-Attn* module of shallow depth can bring selective biases with a task-independent impact on deeper attention modules.

The proposed *Dic-Attn* method, based on feature decomposition and sparse prior, offers a white-box novel strategy for obtaining visual attention, which may be able to be applied to scenarios with more stringent security requirements. Generally, in the visual perceptual system, selective attention is primarily driven by strong stimulus signals and attention to specific regions. Selective attention can be understood from three main aspects: bottom-up voluntary attention, involuntary attention, and "selection history" attention. Involuntary attention, similar to the task-driven model in computer vision, is obtained through top-down rewards. "selection history" refers to the influence of previously attended regions or objects on future attention allocation. Meanwhile, the selective lingering biases have a task-independent impact on the subsequent attention priority. In biology, it means that the nerve has better plasticity. While in the deep neural network, the structure of the multi-layer attention module is not just capable to acquire features with higher semantics. Shallow-depth attention modules capture data prototype features and affect deeper attention modules. It will make models with greater capacity, better generalization, and robustness. The multi-layer attention module structure in the *Dic-Attn* module goes beyond acquiring features with higher semantics. Shallow-depth attention modules capture data prototype features and influence deeper attention modules, resulting in models with increased capacity, better generalization, and improved robustness.

In summary, we demonstrate the effectiveness of the *Dic-Attn* module in capturing visual attention, enhancing visual model performance. We also provide the interpretability for deeper insights into the attention mechanism.

## 4 Experiments

In this section, we evaluate the performance of proposed *Dic-Attn* on 6 benchmarks. Firstly, we conduct ablation experiments on CIFAR-10 dataset, as described in Sec. 4.1. Subsequently, we compare the computational costs. In Sec. 4.2, we evaluate the performance of the *Dic-Attn* module across various visual tasks, such as point cloud classification, image segmentation, and image classification. We compare the accuracy and robustness of the proposed method with several notable attention modules. More experiments and the code are available in the Supplementary Material.

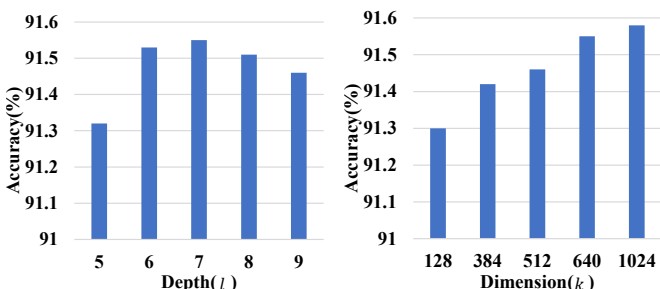

(a) Encoder depth. The deeper the better, but fewer layers can also obtain good accuracy.

(b) Dictionary dimension $k$. An over-completed dictionary can improve performance.

Figure 2: The *Dic-Attn* module ablation experiments (with ViT backbone) on CIFAR-10 dataset.

### 4.1 Ablation Study and Computational Costs

We explore the influence of the hyper-parameters, i.e., the second dimension $k$ of the dictionary and the number $l$ of attention blocks (depth of encoder). Fig. 2 illustrates the significance of selecting appropriate values for $l$ and $k$. A dictionary with higher dimension $k$ creates a comprehensive feature space for images, which enhances the performance of the Vision Transformer. As the depth $l$ increases, the accuracy initially improves but starts to decline after reaching seven in CIFAR-10. The phenomenon of rising first can be attributed to the fact that deeper networks have the capability to capture more intricate relationships between inputs and outputs. However, they are also prone to

overfitting, particularly when the training dataset is relatively small, such as in the case of training from scratch using the CIFAR-10 dataset. Therefore, it is important to carefully consider the trade-offs between depth and complexity when designing a neural network, rather than assuming that deeper is always better.

We evaluate the performance of the *Dic-Attn* module, and compare it with the SA module and notable variants, as presented in Tab. 1. The results show that the proposed *Dic-Attn* module is more lightweight and has a lower computational burden. The total number of parameters (Params.) of the SA module is related to $n$, which is equal to $3n^2$. For our proposed *Dic-Attn* module, the Params. is counted as $k(n + k + 1)$, where Params. of $\mathbf{D}$ is $nk$, Params. of $\mathbf{W_\Phi}$ is $k^2$ and Params. of $W_\mathbf{V}$ is $k$. Since $k = \alpha n$, it can be inferred that $k(n + k + 1)$ is always less than $n^3$ when $\alpha \in (0, \alpha_1) \cap (\alpha_2, +\infty)$. $\alpha_1$ and $\alpha_2$ are solutions of the quadratic equation $n\alpha^2 + (n + 1)\alpha - n^2$ with variable $\alpha$. Note that, it always has real number solutions.

Table 1: Comparison of the efficiency of various attention modules. **Indicators:** Number of parameters (Params.), multiply-accumulates operations (MACs) and training/inference time cost compared to Self-Attention and its variants.

| Methods/Indicators | Params. (M) | MACs. (G) | GPU Train / Inference Time (ms) |
|---|---|---|---|
| **SA** [46] | 1.00 | 292.00 | 242.00 / 82.20 |
| **DA** [5] | 4.82 | 79.50 | 72.60 / 64.40 |
| **A$^2$** [4] | 1.01 | 25.70 | 22.90 / 8.00 |
| **ACF** [59] | 0.75 | 79.5 | 71.00 / 22.60 |
| **HAM** [6] | 0.50 | 17.60 | 15.60 / 7.70 |
| **Dic-Attn** (Ours) | 0.60 | 20.00 | 78.00 / 24.70 |

**Input:** tensor with shape $1 \times 512 \times 128 \times 128$.

## 4.2 Computer Vision Tasks

We evaluate the performances of our proposed *Dic-Attn* module on public image datasets and point cloud datasets, including CIFAR-10, CIFAR-100[20], ModelNet40 [53], ScanObjectNN [45], and ADE20K [63]. Note that, to mitigate the impact of various complex factors that may arise from pre-training and fine-tuning processes, **all models in this paper were trained from scratch, without pre-training, and adopted the same data augmentation method.** Additionally, **unless explicitly stated, the hyperparameter settings remained consistent across all experiments.** More implementation settings are described in detail in the Supplementary Material.

### 4.2.1 Image Classification

**CIFAR-10** dataset is composed of 10 categories 60000 color images with the size of $32 \times 32$. **CIFAR-100** dataset has 100 classes. Both of CIFAR-10 and CIFAR-100 contain 50000 training images and 10000 test images. The number of images in each category is equal, so 6000 images per class in CIFAR-10 and 600 in CIFAR-100, respectively.

The classification results on image datasets are shown in Tab. 2. The image classification accuracy of ViT can be improved from $0.64\%$ to $2.12\%$ by changing the SA module to *Dic-Attn* module.

In Tab. 3, we report the model performance on adversarial robustness under the Fast Gradient Sign Method (FGSM) [7] and Projected Gradient Descent (PGD) [28]. We set the FGSM attack with the perturbation magnitude $\epsilon = \frac{4}{255}$. For PGD, we set $\epsilon = \frac{8}{255}$, iteration numbers $t = 10$, and step size $\alpha = \frac{2}{255}$. We evaluate the model performance by using Attack Accuracy, including the FGSM accuracy and PGD accuracy on classifying images corrupted by adversarial attack FGSM and PGD, respectively. Note that we do not apply adversarial training here, but only use clean samples to train RVT networks [30]. In this case, the proposed *Dic-Attn* module is still competitive.

### 4.2.2 Point Cloud Classification

**ModelNet40** and **ScanObjectNN** datasets are widely used benchmarks for point cloud analysis. ModelNet40 dataset comprises 12311 CAD-generated meshes categorized into 40 classes. Among them, 9843 meshes are used for training, while the remaining 2468 meshes are reserved for testing. ScanObjectNN is a real-world dataset, that contains 15000 objects, divided into 15 and 2902 unique object instances. For all experiments conducted, only the coordinates $xyz$ of objects are utilized. In the case of ModelNet40, all baselines are tested with 1024 input points.

Table 2: Image Classification Top1-Accuracy (%). The overall classification accuracy over all classes is for evaluation metrics.

| Model | Datasets | | #Param(M) |
|---|---|---|---|
| (From Scratch) | CIFAR-10 | CIFAR-100 | |
| ViT (SA) | 90.91 | 66.67 | 22.74 |
| Swin-S (WSA) | 89.74 | 55.82 | 22.25 |
| ViT (Dic-Attn) | 91.55 | 68.79 | 19.33 |

Table 3: Robustness Evaluation Results on CIFAR-100 dataset. Comparison of our approach with backbones and other attention baseline methods, against various adversarial attacks.

| Model | Attacks | | | #Param(M) |
|---|---|---|---|---|
| (From Scrach) | FGSM | PGD | Clean | |
| RVT (SA) | 0.20 | 0.10 | 64.80 | 8.29 |
| RVT (VARS-D) | 10.58 | 3.80 | 62.20 | 7.68 |
| RVT (Dic-Attn) | 10.00 | 4.30 | 55.93 | 7.48 |

The pre-processing of cloud points is performed according to [9]. Note that the point cloud data is encoded as a fixed-shape feature tensor to input the attention module, which preserves and incorporates the original point clouds' position correlation, etc. The classification results on point cloud datasets are shown in Tab. 4. It is evident that our proposed *Dic-Attn* module significantly enhances the classification performance of the Point Cloud Transformer (PCT) model [9]. The average point cloud classification results achieved with our proposed module are highly competitive and demonstrate notable improvement over existing approaches. Our proposed *Dic-Attn* module is able to compute an attention map for the point cloud Transformer. In addition, all these indicate that our proposed *Dic-Attn* module has the advantage of processing complicated data and has the potential to measure the topological attention relationships of the underlying geometric structure.

Table 4: Point Cloud Classification task *Top1-Accuracy* (%). The number of points is set to 1024.

| Model | Datasets | |
|---|---|---|
| (From Scratch) | ModelNet40 | ScanObjectNN |
| PointNet++ [36] | 91.90 | 84.30 |
| PointCNN [22] | 91.70 | 85.50 |
| PCT | 92.95 | 80.60 |
| Point-BERT [57] | 93.19 | 88.10 |
| Point-MAE [35] | 93.80 | 88.30 |
| PCT (Dic-Attn) | 94.60 | 88.96 |
| Point-MAE (Dic-Attn) | 94.46 | 89.41 |

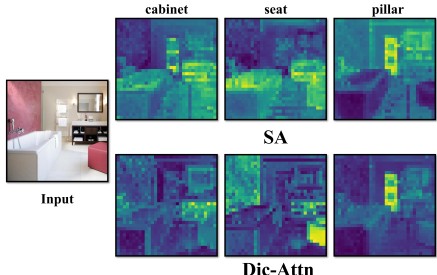

Figure 3: The comparison of attention maps generated by SA module and *Dic-Attn* module (with backbone Segmenter, experiment on ADE20K dataset).

### 4.2.3 Image Segmentation

**ADE20K** dataset [63] contains a total of 150 semantic categories, and can be used for scene perception, parsing, segmentation, multi-object recognition, and semantic understanding, which is one of the most challenging semantic segmentation datasets. The training set contains 20210 images. There are about 2000 and 3352 images in the validation and test set, respectively.

The performance of the *Dic-Attn* module is verified in the semantic segmentation task with the backbone Segmenter [40], which is a competitive and representative transformer-based semantic segmentation model. We train Segmenter (SA) and Segmenter (Dic-Attn) from scratch on ADE20K dataset. Note that, Segmenter (Dic-Attn) denotes the proposed *Dic-Attn* module replacing the original SA module in the Segmenter backbone. The mean Intersection over Union (mIoU) and the mean Pixel Accuracy (mPA) of Segmenter (Dic-Attn) are 1.00% and 1.25% higher than that of Segmenter (SA), respectively.

Furthermore, the two image examples and their corresponding segmentation attention map results, as shown in Fig. 3, intuitively indicate that our proposed *Dic-Attn* module highly promotes the ability of the segmentation model to distinguish objects. The learned attention maps are also more similar to human visual attention, which focus on referred objects, e.g., building, sky, door, cabinet, etc. The learned dictionary in the *Dic-Attn* module contains more sufficient features, including the underlying nonlinear structure. All results above indicate that by fully extracting attention information in both the channel domain and spatial domain of feature space, the attention maps for different labeled objects can be reconstructed more accurately.

# 5 Conclusions

In this work, a novel dictionary learning-based attention module, namely *Dic-Attn*, was proposed. *Dic-Attn* takes the advantage of dictionary learning mechanism, enabling effective exploration of the underlying nonlinear structure information. By dynamically selecting learned dictionary atoms and sparse representations through spatial and channel transformations, multi-layer *Dic-Attn* modules achieve the accurate reconstruction of attention maps and allow for the extraction of features at different semantic levels, enhancing the comprehensive representation of visual information. Experimental results in various visual tasks and vision models showed that the *Dic-Attn* module can seamlessly integrate into various vision networks, delivering competitive performance comparable to state-of-the-art methods. The *Dic-Attn* module is also a promising solution with competitive performance, compatibility, and computational efficiency. Further research will be devoted to the robustness of the module and its applications in more tasks, e.g., temporal prediction tasks in scientific and engineering fields. Conducting further investigation and validation of the diverse levels of semantics exhibited by dictionaries learned at different depths in natural language processing scenarios would also be valuable.

## Acknowledgements

This work was supported by the National Natural Science Foundation of China (No.42130112, No.41901335), KartoBit Research Network (No.KRN2201CA), and the Shanghai Municipality's "S&T Innovation Action Plan" for the year 2023, Shanghai Natural Science Fund, General Program (No.23ZR1419300), Shanghai Trusted Industry Internet Software Collaborative Innovation Center, "Digital Silk Road" Shanghai International Joint Lab of Trustworthy Intelligent Software (No.22510750100).

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
