# Supplemental Material for
# Learning Dictionary for Visual Attention

**Yingjie Liu**[1]   **Xuan Liu**[2]   **Hui Yu**[2]   **Xuan Tang**[1]   **Xian Wei**[1*]

[1]East China Normal University
[2]FJIRSM, Chinese Academy of Sciences

This supplementary material provides additional information for this research. First, we explain the configuration employed across all experiments. Then, we present further experimental explanations, including heatmaps generated from different depths and the visualization of Attention Output. Finally, we give theoretical derivations and some details related to the dictionary learning problem.

## 0.1   Experiments Configures

Table 1: Implementation Settings

| Configuration | Value |
|---|---|
| data_augmentation | padding, crop, normalize, random horizontal flip |
| image_size | $32 \times 32$ |
| optimizer | AdamW |
| learning_rate | 1e-4 |
| warm_up_epoch | 5 |
| batch_size | 32 |
| training_epoch | 300 |
| embedding_dimension | 768 |
| encoder_depth | 7 |
| patch_size | 4 |

As Tab. 1 shows, we maintained a consistent set of configurations for the neural network in all image classification experiments. This approach ensured the comparability and reproducibility of the results. We applied specific data augmentation techniques, including padding, crop, normalize, and random horizontal flip. All input images were resized to a $32 \times 32$ and with a fixed batch size 32 in CIFAR-10 and CIFAR-100. We employed a specific optimizer AdamW with a warm-up strategy. We performed a fixed number of 300 of training iterations. All Vision Transformer backbones have the same encoder depth, patch size, and embedding dimension.

## 0.2   Visualization of Attention Outputs

Attention maps (heatmaps) help understand how the model attends to different regions of the input data and highlight the areas of focus. We generated attention maps to visualize the activation patterns of the neural network at various depths. As the attention maps generated by our proposed *Dic-Attn* module, shown in Fig. 1, it can be seen that the *Dic-Attn* modules of different layers in Vision Transformer generate different attention maps that provide attention to different key parts of the image.

---

[*]Corresponding author: `xwei@sei.ecnu.edu.cn`

37th Conference on Neural Information Processing Systems (NeurIPS 2023).

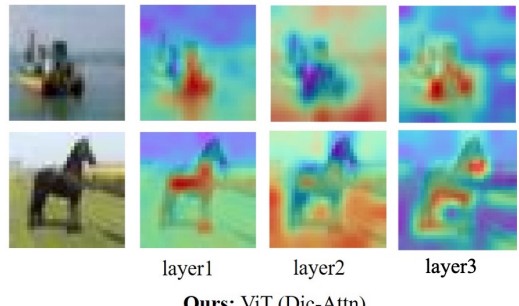

layer1          layer2          layer3

**Ours:** ViT (Dic-Attn)

Figure 1: Attention maps (heatmaps) are generated by the *Dic-Attn* modules of different layers.

We learn the dictionary for obtaining visual attention. Here we visualize the sparse reconstruction from a learned dictionary. We can see that reconstructed attention maps (*Attention Outputs*) capture various information about images, including either major objects or irrelevant background regions.

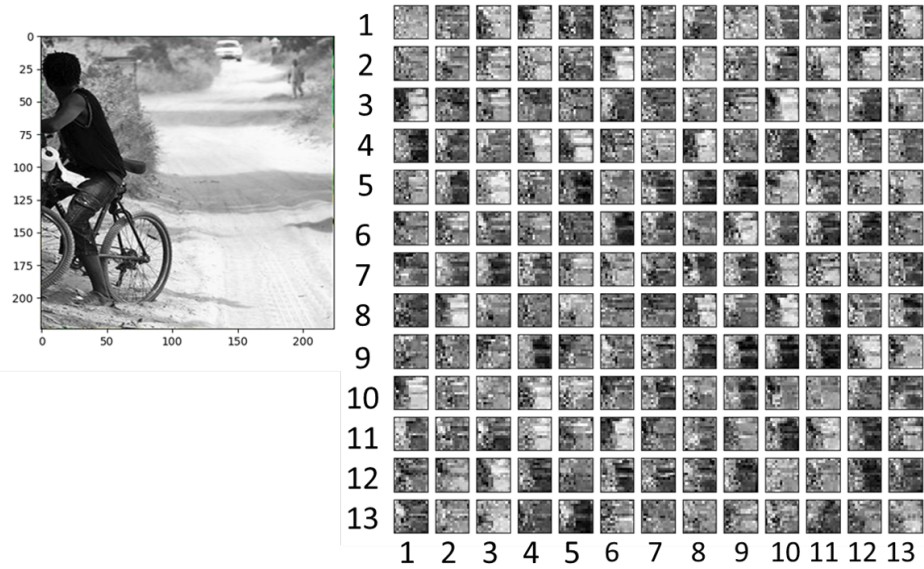

Figure 2: Visualization of the *Attention Outputs*. We visualize the attention map of ViT(Dic-Attn).

## 0.3 More Visual Experiments

**Image Classification**.

We have conducted additional experiments on the ImageNet-1k dataset for image classification. We use "Swin-T(WSA)" and "Swin-T(Dic-Attn)" to denote the Swin Transformer-Tiny model and Swin Transformer-Tiny model using the Windows Self-Attention module the proposed dictionary learning-based visual attention(Dic-Attn) module, respectively. Specifically, the added results of the classification task on ImageNet-1K are shown as follows:

**Denoising**.

We have conducted experiments on the SIDD dataset for image denoising. The results of these experiments show that the transformer augmented with our proposed module outperforms the baseline in denoising performance.

Table 2: Additional classification results on ImageNet-1K. Models are trained from scratch on ImageNet-1K. The input images are all resized to $224^2$. Our proposed Dic-Attn module brings up 1.37% gains in Top-1 accuracy compared with backbone Swin-T(WSA) at the beginning of training (50 epochs).

| Methods | Resolution | Acc (50 Epochs) | Acc |
|---|---|---|---|
| Swin-T(WSA) | $224 \times 224$ | 70.29 | 81.23 |
| win-T(Dic-Attb) | $224 \times 224$ | 71.66 | 81.29 |

Table 3: Additional image denoising results on SIDD.

| Methods | PSNR | SSIM |
|---|---|---|
| Restormer(SA) | 39.53 | 0.960 |
| Restormer(Dic-Attn) | 39.54 | 0.958 |

## 0.4 Theoretical Derivation and Details Related to Dictionary Learning Problem

We assume and model the selective attention problem as solving a linear inverse problem $f_{rec}$ in which preprocessed visual features are mapped non-linearly to generate an attention map, i.e.,

$$x = \textbf{Attention Output} + \epsilon$$
$$\textbf{Attention Output} = \mathbf{D}\phi \tag{1}$$

where $\epsilon$ denotes irrelevant features and noise, and **Attention Output** is the generated attention map, which is reconstructed from a dictionary and the corresponding sparse codes. In this formulation, determining $\phi$ from the measurements $x$ is the famous linear inverse problem. In other word, the dictionary $\mathbf{D}$ is the measurement system modeling the selection process. Visual attention can be further obtained from this dictionary.

The sparse representation $\phi$ of $x$ is formulated in sparse-constrained minimization problem

$$\phi^* := \underset{\phi}{\arg\min}\, g(\phi),$$
$$s.t. \mathbf{D} \in \mathcal{S}(n,k), ||x - \mathbf{D}\phi||_2 \leq \epsilon, 0 \leq p \leq 1, \tag{2}$$

where $g(\phi)$ is the sparse regularization term, and $D(m,k)$ is predefined admissible set of solutions for $\mathbf{D}$,such as wavelet bases or sphere manifold adopted in this work. Traditional, $g(\phi) = \|\phi\|_0$. When $p = 0$, $\|\phi\|_0$ counts the number of nonzero terms in $\phi$ and leads to sparse coding problem as a general NP-hard. When $p = 1$, Eq. (2) can be relaxed to a convex optimization problem. Elastic net [2] now is an outstanding and popular convex relaxation way. Hence in this work, $g(\phi) = \beta \|\phi\|_1 + \frac{\lambda}{2} \|\phi\|_2$. Hyper-parameters $\beta$ and $\lambda$ are introduced to control the sparsity or change the proportion of Laplace distribution of mixed prior distributions of sparse codes.

Let us assume that the probability of $(x, \textbf{Attention Output})$ admits a continuous density with compact support, and final object function $F_{loss}(x, \Theta, \textbf{Attention Output}))$ is twice differentiable, the sparse coding problem is differentiable. Note that $\Theta$ denotes the set of parameters of neural networks. Hence, the partial differential w.r.t. $\mathbf{D}$ can be expressed as $\mathbb{E}_{x, \textbf{Attention Output}})[-\mathbf{D}\phi^*\phi_\Lambda^{*T} + (x - \mathbf{D}\phi_\Lambda^*)\phi^*)]$, and $\phi$ is a vector that depend on $x, \mathbf{D}, \Theta$. Let the derivative of the dictionary loss w.r.t $\phi$ be zero,

$$\frac{\partial Tr((x - \mathbf{D}\phi)^T(x - \mathbf{D}\phi))}{\partial \mathbf{D}} = 0, \tag{3}$$

we have the closed-form solution of $\phi$,

$$\phi_i^* = \left(\mathbf{D}_\Lambda^T \mathbf{D}_\Lambda + \lambda \mathbf{I}\right)^{-1} \left(\mathbf{D}_\Lambda^T \mathbf{x}_i + \beta sign(\phi_i^*)\right), \tag{4}$$

where $\Lambda := \{i \in \{1,...,s\}|\phi_{ij}^* \neq 0\}$ denotes the set of indexes of the non-zero entries of the solution $\phi_i^* = [\phi_{i1}^*, .., \phi_{is}^*]$. $\mathbf{D}_\Lambda$ is the subset of the dictionary in which the indexes of atoms fall into $\Lambda$. $sign(\phi_i^*)$ carries the signs of $\phi_i^*$. More investigation of the differentiability of the sparse

representation in the dictionary can be also found in [4]. So far, we can easily compute $\phi_i^*$ and update dictionary $\mathbf{D}$ wtih mini-batches and stochastic gradient decent algorithm [1, 3, 4],

$$\mathbf{D}^* = \mathbf{Proj}_{orth}(\mathbf{D} - \delta\nabla_{\mathbf{D}}F_{loss}(x, \mathbf{Attention\ Output})), \tag{5}$$

where $Proj_{orth}$ denotes the orthogonal projector onto $\mathcal{S}(n, k)$. $\mathbf{D}^*$ denotes the optimal dictionary at the current iteration, In this paper, the dictionary $\mathbf{D}$ can be initialized in an unsupervised manner. Also, the updated gradient of the dictionary is considered to be provided by the visual data and downstream task, since the back-propagation gradient $\nabla_{\mathbf{D}}F_{visTask}$ consists of two parts, $\frac{\partial F_{loss}}{\partial \phi}$ and $\frac{\partial f_{rec}}{\partial \phi}$.

Finally, the output Attention Maps are obtained through the reconstruction of the re-weighted dictionary and transformed sparse representations. The diagonal transformed matrix and transformed matrix are specifically designed to re-weight the dictionary and transform its corresponding sparse representations, respectively. $\mathbf{W_D}$ operates in a vector-wise manner, where each element in the diagonal serves as a weight for the dictionary columns, i.e., atoms. On the other hand, $\mathbf{W_\Phi}$ implements the sparse encoding of individual elements. These two parameters are updated via the backpropagation strategy and are thus driven by the task's final objective function.