# OpenReview forum: "Learning Dictionary for Visual Attention"
_NeurIPS.cc/2023/Conference — NeurIPS 2023 poster_

### Official Review · Reviewer_q2Dg · 2023-07-06

**Soundness:** 2 fair
**Presentation:** 2 fair
**Contribution:** 1 poor
**Rating:** 5
**Confidence:** 4

**Summary:**

This paper presents a new architecture that can be applied for various tasks, including image classification, point cloud classification and image segmentation. The key idea is to leverage a learnable dictionary module to replace the attention model in the transformer architecture. The model is able to achieve compelling results on multiple tasks with the low computational cost, including the fewer parameters, less GPU training time. Experimental results are conducted on multiple vision tasks and achieve compelling results.

**Strengths:**

### improved results over baseline models
- Based on the quantitative results shown in Tables 2, 3 and 4, the proposed model outperforms the existing models on all tasks.



**Weaknesses:**

### (Major) Confusion on method
- It took me a long time to actually get the main message of this paper, which is hidden in a bunch of overwhelming technical details and verbose writing.
- Most importantly, the dictionary learning directly learn the sparse representation for the attention $A$. However, this is just a sparse representation, and I cannot get the attention similarity between tokens of different grid features.
- The equation $D=[d_1^c,d_2^c,\cdots,d_k^c]=[d_1^r, d_2^r,\cdots,d_s^r]^T\in R^{k\times s}$ and $\Phi=[\phi_1^c,\phi_2^c,\cdots,\phi_k^c]=[\psi_1^r, \psi_2^r,\cdots,\psi_s^r]^T\in R^{n\times s}$ are confused, which is different to the dictionary $D\in R^{n\times s}$and sparse representation $\Phi\in R^{k\times s}$ defined before.
- The equation is also confused. $x_{1*}$ denotes one grid feature with different dimension value, but every channel is represented by different code $\phi$, which is hard to understand for the embedding.
- The key optimisation function is also unclearly. The dimension of $\mathbf{D}$ and $\mathbf{\Phi}$
is different. How could they calculate the reconstruction in equation (6)?

### (Major) More results should have been expected
- I expected to see more segmentation results, but only the compared attention map is provided in Figure 3, and no results in supplemental material either.
- A quantitative comparison is expected for the image segmentation task.
- It would be helpful to provide some visual examples of the segmentation map in the main paper.

### (Minor) Presentation
- For figure 2, the paper claimed "the deeper the better", but it is not true from the figure visual results. It would be stronger to provide the accuracy analysis for the results.

**Questions:**

Apart from a few questions that need clarifications listed above, the only additional results I might suggest is to also perform inference time reported as in some related works for a slightly fairer comparison.

**Limitations:**

There are no discussion or broader societal imparts discussed in the main paper.

---

> ### Author Rebuttal · Authors · 2023-08-10
>
> We sincerely thank for your valuable suggestions on this paper. The following responses might help address your questions and concerns about this paper:
>
> #### 1. **Weakness 1. (Major) Confusion on method...**
>
> **Q(1)** "...The key motivation seem..."
>
>    **Answer to Q(1):** Thank you for your careful review and feedback. In the **revised version**, we further improve the clarity and organization of the paper to ensure that the main message is more easily accessible to readers.
>
>    The primary focus of our paper is on introducing the concept of dictionary learning-based visual attention module and showcasing its potential for capturing visual attention. The attention mechanism is one of the key structures of the very popular high-performance transformer network. Recalling that in human visual perception, a large number of studies have shown (some references are cited in the  Introduction and Section 3.3.2) the primary visual perception cortical signal encoding method and Dictionary learning is similar, which is also closely related to human visual attention. Inspired by this, we attempt to propose a dictionary learning-based visual attention (Dic-Attn) module in machine vision, exploiting the potential nonlinear structural information disentanglement ability from dictionary learning, and exploring its intuitive way to exploit discriminative information and visual attention.
>
>    **Q(2)** "...I cannot get attention similarity between tokens of different grid features..."
>
>    **Answer to Q(2):**
>    The sparse representation serves as the coordinates of the dictionary. These sparse representations indicate the importance or relevance of each atom in the dictionary for reconstructing the input features. Importantly, the attention similarity between different tokens or grid features is inherently encoded within the dictionary and the sparse representation.
>    The diagonal transformed matrix $\mathbf{W_D}$ and transformed matrix $\mathbf{W_{\phi}}$  are specifically designed to re-weight the dictionary and transform its corresponding sparse representations, respectively. Moreover, $\mathbf{W_D}$ operates in a vector-wise manner, where each element in the diagonal serves as a weight for the dictionary columns, i.e., atoms. On the other hand, $\mathbf{W_{\phi}}$ implements the sparse encoding of individual elements.
>    These transformation parameters are driven by the task objective function and are updated through the backpropagation process.
>
> **Q(3)** "The equations are confused..."
>
>    **Answer to Q(3):** We apologize for the confusion and appreciate your correction. In the **revised version**, we have corrected these typos:
>
>    $D = [d^c_1  d^c_2  \cdots  d^c_k]= [d^r_1 d^r_2 \cdots d^r_n ]^T\in R^{n\times k},$
>
>    $\Phi = [\Phi_1  \Phi_2  \cdots  \Phi_b]\in R^{b\times k\times s},$
>
>    $\Phi_i = [\phi_1  \phi_2  \cdots  \phi_s]= [\psi_1 \psi_2 \cdots \psi_k ]^T\in R^{k \times s}.$
>
> In our proposed method, we aim to capture the critical part of visual data by utilizing dictionary learning. According to experience, we build the dictionary as $n*k$, instead of building a dictionary at the fine-grained level of batch-level. The results also show that dictionary as a feature base is complete enough and discriminative. Hence, given the input $X\in R^{b\times n \times s}$, sparse representations $\Phi \in R^{b\times k\times s}$ will be seen as its sparse embedding.
>
> **Q(4)** "The key optimization function... calculate the reconstruction in equation (6)"
>
> **Answer to Q(4):**  We appreciate for pointing out the unclear optimization function and the refactoring of the computation in Equation 6.
>
> Firstly, we clarify how the optimization process works in Algorithm 1 and the key optimization function for dictionary learning. Due to space limitations, please refer to our response to Weakness 1 from Reviewer nmhX. Additionally, in the revised version, we provide a more detailed explanation of the key optimization function for dictionary learning.
>
> After correcting Typo, we illustrate how could $D$ and $\Phi$ calculate the reconstruction in equation 6.
> Therefore, let $X_i=[x_1, \cdots, x_s]\in R^{n\times s}$ denotes the input, $D\in R^{n\times k}$ represents the dictionary, $\phi$ represents the sparse coding vector, $\Phi_i = [\phi^c_1  \phi^c_2  \cdots  \phi^c_k]\in R^{k\times s}$ . The non-linear scaling function NL(·) in equation 6 and equation 4 do not change the dimension of sparse representation $\Phi$. Therefore, it is able to calculate the reconstruction and output the attention map, i.e., $R^{n\times s} \rightarrow R^{n\times k} \times R^{k\times s} \rightarrow R^{n\times s}$.
>
> #### 2. **Weakness 2 / Question.  "(Major)More results should have been expected...; "...perform inference time reported..."**
>
> **Answer.** Specifically, we add more experiments to verify our proposed module explicitly. We now place added experimental results in the rebuttal box at the top. The added results of inference time are also shown in the rebuttal box at the top. We appreciate your suggestion and hope that the revised version will better address your concerns.
>
> #### 3. **Weakness 3. (Minor) Presentation...**"
>
> **Answer.** Thank you for your careful review and feedback. Based on the results, it can be observed that as the depth increases, the accuracy initially improves, but after reaching a certain depth, the accuracy starts to decline. The phenomenon of rising first can be attributed to the fact that deeper networks have the capability to capture more intricate relationships between inputs and outputs. However, they are also prone to overfitting, particularly when the training dataset is relatively small, such as in the case of training from scratch using the CIFAR-10 dataset.
> 	Therefore, it is important to carefully consider the trade-offs between depth and complexity when designing a neural network, rather than assuming that deeper is always better.

---

> > ### Comment · Reviewer_q2Dg · 2023-08-17
> > **Thanks for the rebuttal**
> >
> > Thanks for providing these detailed responses. All my concerns have been addressed. The rebuttal provides a good analysis in terms of the proposed methods and more results. This new learnable dictionary module is meaningful for the community. Therefore, I raise my score for accept.

---

> > > ### Author Response · Authors · 2023-08-18
> > >
> > > We are glad that our response has addressed your concern and thank you for raising your score for our paper! The temporary rating we obtained is borderline accept rating (5) in the review system. We will carefully revise the manuscript and include the experiments to the final version. Appreciate again for your time and look forward to your considering raising your score to a higher accept rating.

---

### Official Review · Reviewer_1dqN · 2023-07-06

**Soundness:** 2 fair
**Presentation:** 2 fair
**Contribution:** 2 fair
**Rating:** 6
**Confidence:** 4

**Summary:**

This paper is about a new attention module called Dic-Attn, which is based on dictionary learning and sparse coding in the human visual perception system. The module can extract nonlinear structural information in visual data and reconstruct attention maps. The paper emphasizes the potential of leveraging sparsity in attention algorithms and addresses the efficiency and effectiveness of current attention methods. Authors also conduct experiments on image classification, point cloud classification.


**Strengths:**

1. The idea is novel.This paper proposes a novel attention module, Dic-Attn, which combines dictionary learning and attention mechanism to effectively explore the underlying nonlinear structure information and enhance the comprehensive representation of visual information.

2. The performance on point cloud classification is significant, which demonstrates the effectiveness of proposed method.

3. The writing is esay to follow and visualization is reasonable.



**Weaknesses:**

1. Experiments on CIFAR10 for image classification are not convincing. Please conduct experiments on ImageNet dataset to compare swin transformer.  Besides, showing the result on ADE20K dataset.

**Questions:**

Seeing weaknesses

**Limitations:**

Seeing weaknesses

---

> ### Author Rebuttal · Authors · 2023-08-10
>
> We appreciate your careful review and feedback on this paper.
> The following responses might help address your questions about this paper:
>
> **Question & Weakness .** "Experiments on CIFAR10 for image classification are not convincing. Please conduct experiments on ImageNet dataset to compare Swin transformer. Besides, showing the result on ADE20K dataset."
>
> **Answer.** We highly appreciate your valuable advice. In the revision, we have illustrated the additional experimental results.
>
>    (1) We use "Swin-T (WSA)" and "Swin-T (Dic-Attn)" to denote the Swin Transformer-Tiny model and Swin Transformer-Tiny model using the windows Self-Attention (WSA) module and the proposed dictionary learning-based visual attention (Dic-Attn) module, respectively.
>
>    (2) Specifically, the added Accuracy (Acc), Mean Intersection over Union (mIOU) / Mean Accuracy (mAcc) results of **the classification task on ImageNet-1K** and **the part segmentation task on ADE20K** are shown as follows:
>
>   **Table 1.**  Additional classification results on ImageNet-1K.
>
> | Methods          |   Dataset   |   Resolution    | Acc (50 Epochs) |  Acc  |
> | ---------------- |:-----------:|:---------------:|:---------------:|:-----:|
> | Swin-T (WSA)      | ImageNet-1K | $224\times 224$ |      70.29      | 81.23 |
> | Swin-T (Dic-Attn) | ImageNet-1K | $224\times 224$ |      71.66      | 81.29 |
>
>    Swin-T (WSA) and Swin-T (Dic-Attn) are trained from scratch on the ImageNet-1K dataset on four GeForce RTX 4090 GPUs for 300 epochs. The input images are all resized to $224^2$, and the batch size is 256. The training process for each epoch is expected to take approximately one to two hours. Our proposed Dic-Attn module brings up 1.37% gains in Top-1 accuracy compared with Swin-T(WSA) at the beginning of training (50 epochs).
>
> #### **Table 2.  Additional segmentation results on ADE20K.** Backbones are trained from scratch on ImageNet-22K. ViT-Adapter-L applies UperNet framework.
>
> | Methods                |Backbone| mIOU  | mAcc  |
> | ---------------------- |:-----:|:-----:|:-----:|
> | Segmenter-B          |ViT(SA)| 49.20 | 59.32|
> | ViT-Adapter-L       |Beit(SA)| 56.80 | 69.56 |
> | Segmenter-B    |ViT(Dic-Attn)| 50.20 | 60.57  |
> | ViT-Adapter-L |Beit(Dic-Attn)| 56.93  | 69.78 |
>
> More experiments are still on evaluated, and the results will be published as soon as possible and added to the **revised Manuscript and updated Supplementary Material**.

---

> > ### Comment · Reviewer_1dqN · 2023-08-12
> > **Response to rebuttal**
> >
> > Thanks to the efforts in the rebuttal stage, I appreciate the author's experiments and response. I am willing to raise my score to 6 (weak accept). Please make sure that all experiments will be included to the final version.

---

> > > ### Author Response · Authors · 2023-08-12
> > >
> > > We are glad that our response has addressed your concern and thank you for raising your score for our paper! We will carefully revise the manuscript and include the experiments to the final version.

---

### Official Review · Reviewer_Vec1 · 2023-07-06

**Soundness:** 2 fair
**Presentation:** 2 fair
**Contribution:** 2 fair
**Rating:** 5
**Confidence:** 4

**Summary:**

This paper proposed a novel dictionary learning-based attention (Dic-Attn) module, The proposed Dic-Attn module can be plug-and-play and stacked layer by layer to form a deep attention encoder. Extensive experimental results on computer vision tasks, e.g., image classification and point cloud classification, demonstrate the performance of the proposed method.

**Strengths:**

This paper proposed a novel dictionary learning-based attention (Dic-Attn) module, The proposed Dic-Attn module can be plug-and-play and stacked layer by layer to form a deep attention encoder. This is an interesting idea, might be worth to keep on digging in.

**Weaknesses:**

1. Writing: This is not a well-organized paper. It is more like an application paper that pursues SOTA results rather than introducing an interpretable layer.  If I understand it correctly, this paper does not focus on any analysis of the sparse dictionary learning but applies it to the transformer. If so, I would expect some insights, e.g the self-attention module could be regarded as sparse dictionary learning under some conditions and offer the proof.  Then, the paper should squeeze the length of sections 3.1 and 3.2, and leave more space to discuss or explain the insights of the method.

2. Experiments: As mentioned above, this paper is an application of sparse dictionary learning in the transformer, so readers would expect some explainable experiment results or better performance than the baseline. Just experimenting on small datasets like CIFAR is not convincing when we know that the transformer was commonly used in large datasets like ImageNet.  We know that sparse dictionary learning algorithms usually be applied in robustness or denoising tasks and could achieve good results. And the author should conduct some experiments that a transformer weapon with the proposed module should be significantly better than the baseline on robustness or denoising tasks.

**Questions:**

please refer to weakness.

**Limitations:**

please refer to weakness.

---

> ### Author Rebuttal · Authors · 2023-08-10
>
> We appreciate your careful comments and constructive suggestions. The followings are detailed responses to your questions/concerns:
> 1. **Weakness 1.** Writing: This is not a well-organized paper. It is more like an application paper that pursues SOTA results rather than introducing an interpretable layer. If I understand it correctly, this paper does not focus on any analysis of sparse dictionary learning but applies it to the transformer. If so, I would expect some insights, e.g the self-attention module could be regarded as sparse dictionary learning under some conditions and offer the proof. Then, the paper should squeeze the length of sections 3.1 and 3.2, and leave more space to discuss or explain the insights of the method.
>
>     **Answer.** We appreciate your thoughtful comments on the organization and focus of our paper. In the **revised version**, we strike a better balance between presenting SOTA results and providing a clear explanation of the method.
>
>     (1) Section 3.1 discusses the summarized CV attention form, which is also the foundation of our proposed Dic-Attn module. Section 3.2 introduces the dictionary learning problem and derives the close solution of the sparse representation. With these two crucial parts at hand, we propose the concept of dictionary learning-based attention within the context of transformer models. We analyzed the process of data processing in the model and discussed the way the proposed module extracts attention. At the same time, due to the correlation between dictionary learning and biological visual perception, we reviewed some biological-related viewpoints as evidence for the discussion.
>
>     (2) While it is true that our paper primarily focuses on achieving state-of-the-art results, it is not solely an application paper. We also analysis the influencee of the encoder depth (model perspective), dictionary dimension, and the sparsity regularization coefficient (Dic-Attn module perspective). Further, we discuss the number of parameters and computational burden. Experimental results in image classification (CIFAR-10/100, ImageNet-1K), robust evaluation (CIFAR-10), real image denoise (SIDD), image segmentation (ADE20K) and Point Cloud Classification (ModelNet40, ScanObjectNN) showcasing the potential of dictionary learning-based visual attention module for capturing visual attention. The potential non-linear structural information disentanglement ability of dictionary learning can improve performance during segmentation, which has been validated in the comparison of attention maps within the ADE20K experiments.
>
>     (3) Regarding your suggestion to explore the self-attention module as sparse dictionary learning under certain conditions, in section 3.1 of our current paper, we provide a general attention form that also includes the self-attention mechanism. Although both have operations such as inner product, the operations inside the module are completely different, as can be seen by comparing Equation 1 and Equation 4. The most important step in obtaining attention maps for Dic-Attn is nonlinear decomposition and reconstruction, while the process of SA obtaining query, key, and value is a linear transformation. However, we agree that such analysis can provide valuable insights, and we will consider incorporating this aspect into future research or expanding our work to explore the connection between self-attention and sparse dictionary learning.
>
> 	Thank you once again for your valuable feedback and constructive suggestions!
>
>      &emsp;
>
> 2. **Weakness 2**. Experiments: "As mentioned above, this paper is an application of ..."
>
>     **Answer.** We now place more experimental results in the rebuttal box at the top.  We appreciate your valuable feedback, as it helps us improve the comprehensiveness and applicability of our experimental evaluation. More experiments are still on evaluated, and the results will be published as soon as possible and added to the **revised Manuscript and updated Supplementary Material**.

---

> > ### Comment · Reviewer_Vec1 · 2023-08-17
> > **Feedback**
> >
> > I appreciate the solid feedback from the authors and apologize for missing the robustness results. However, the denoising results did not appear in the original manuscript after a double-check.
> >
> > Going back to the first question, I think the authors might not really get my question, The purpose of interpolation should serve to the follow-up benefits, like faster, better performance, robustness, denoising, easy training, or something else related. I would hold a tolerant attitude towards the follow-up benefits if I reviewed this paper one or two years ago (since works like VARS already published). However, from the perspective of diversity and novelty exploration, your work actually have some differences with those sparse dictionary learning works. And I also agree novelty needs time (time of engineering) to match its real value. So, could you please justify the meaning of your work for the follow-up benefits, or some new insight that contribute the Transformer/Sparse dictionary learning communities?
> >
> > The justification could be the discussion of future work about the reasonable potential of your method. For example, we know the sparse dictionary learning holds two main properties, 1) linear inverse, 2) sparsity, any follow-up benefits of your method could based on these two properties?
> >
> > I will consider to change my score based on your response.

---

> > > ### Author Response · Authors · 2023-08-18
> > >
> > > We sincerely thank you for careful review and thoughtful feedback. We will carefully revise the original manuscript and ensure that the necessary information is included to the final version! In response to the first question, we explain from two aspects:
> > >
> > > - **Our exploration:**
> > >
> > >   - Upon studying and summarizing the general paradigm of computer vision attention and the concerns of biological vision in Section 3, we discovered that the crucial step in both our proposed module and existing successful computer vision attention modules involves **nonlinear decomposition and reconstruction**.
> > >      As highlighted by the reviewer, dictionary learning holds two key properties: **1) linear inverse, and 2) sparsity**/
> > >
> > >   - Specifically, dictionary learning aims to learn a set of atoms such that a given signal can be well approximated by a sparse linear combination of these atoms. Dictionary learning process can be viewed as a form of matrix decomposition with a sparse constraint, which is a NP-hard optimization problem. VAR formulated this optimization process as an ODE recurrent attractor network that captures attention. It also demonstrated that sparsity leads to attention emergence and improves model robustness. **In our paper, we propose an alternative attention method and an implementation of dictionary learning based attention module.** We employ Elastic Net regularization to enforce sparsity in the dictionary learning and sparse coding problem. The use of Elastic Net regularization transforms the problem from NP-hard to a convex optimization problem, for which a closed-form solution for the sparse representation exists.
> > >
> > >   - Our proposed module emphasizes the significance of the sparse constraint in enhancing the discriminative power of the learned dictionary, similar to the objective of traditional dictionary learning in solving visual tasks. **It is worth noting that numerous visual processing problems can be formulated as inverse problems, the two key properties possessed by dictionary learning solve them well.** Hence our proposed module reveals great potential in disentangling non-linear structural information. As the attention module plays a crucial role in Vision Transformers, models equipped with our proposed Dic-Attn module exhibit  demonstrate accelerated training convergence and improved performance in inverse problems such as denoising and robustness, as well as other tasks including segmentation, and classification.
> > >
> > > - **Follow-up benefits and possible future research:**
> > >
> > >   - **Conducting further investigation and validation of the diverse levels of semantics exhibited by dictionaries learned at different depths in NLP scenarios** would be valuable.
> > >
> > >   - Transformer/Sparse dictionary learning communities can further **investigate the trade-off between sparse reconstruction accuracy and the performance (robust, accuracy, generalization, ...) of the model**. Specifically, in this article, we achieved complete sparse reconstruction in the proposed module, and the attention map is subsequently derived from the task-driven selection matrix. Consequently, considering the large number of parameters of Transformer model, we can explore **the possibility of intentionally reducing the accuracy of sparse reconstruction within the attention module**.
> > >
> > >   - Additionally, when the learned dictionary as a feature base, the advantages of **low computation complexity and cost** overhead in sparse coding can facilitate future edge deployment and training. These topics have never been raised or discussed in related papers for the Transformer + dictionary learning communities.
> > >
> > > Thanks again for your constructive comments! We truly appreciate your time and look forward to your considering raising your rating to a higher accept rating!

---

> > > > ### Comment · Reviewer_Vec1 · 2023-08-19
> > > > **Score Raised**
> > > >
> > > > I appreciated the authors' careful and serious feedback. This time, the authors got my point and I raised my score to 5.
> > > >
> > > > I encourage the authors to keep on digging into this interesting research direction (the application of sparse dictionary learning in Transformer) and then make the future work you mentioned in the feedback come true. Especially, applying it to the NLP community could be a good first step.
> > > >
> > > > Good luck.

---

> > > > > ### Author Response · Authors · 2023-08-19
> > > > >
> > > > > We appreciate again your constructive suggestions and raising your score for our paper! We will carefully revise the manuscript.

---

### Official Review · Reviewer_nmhX · 2023-07-07

**Soundness:** 3 good
**Presentation:** 3 good
**Contribution:** 3 good
**Rating:** 5
**Confidence:** 3

**Summary:**

This paper introduces a new attention mechanism, dictionary learning-based attention (Dic-Attn), to replace existing attention modules (e.g., self-attention) in deep networks (e.g. Vision Transformer, ViT). The proposed Dic-Attn comes from the combination of dictionary learning and sparse coding, and sparse visual attention. The Dic-Attn module is used to replace self-attention in transformer models for image classification, point cloud classification, and image segmentation. Results show the Dic-Attn performs better than self-attention on these tasks.

**Strengths:**

1. This paper presents a very interesting idea to combine dictionary learning and sparse coding with attention mechanisms. Attention is proposed with the hypothesis that not all features are equal and we should only pay attention to important ones. The spirit is similar to sparse representation learning. The proposed Dic-Attn combines them in an elegant way.
2. The proposed Dic-Attn plugged into the existing models is shown to outperform self-attention for various computer vision tasks: image classification, point cloud classification, and semantic segmentation, with noticeable improvements.
3. The influence of hyper-parameters, dimension k and attention block number l, is studied and the results provide some insight of how they may affect the model performance. Essentially, larger k and larger l leads to better performance.


**Weaknesses:**

1. How does the backward process work in Algorithm 1? The sparse representation \phi in Eq. 3 is not differentiable.
2. What are the mIoU values of the proposed method and the counterpart using self-attention on the ADE20K dataset for semantic segmentation?
3. Since the paper proposes to learn sparse representation using dictionary learning-based attention, some example sparse representations should be shown in order to verify the effectiveness of the proposed method. While Fig. 3 visualized some attention maps for certain categories, the sparsity of the representation is still unknown.


**Questions:**

Please address the concerns above.

**Limitations:**

The authors did not discuss the limitations of this work.

---

> ### Author Rebuttal · Authors · 2023-08-10
>
> Thank you for your careful review and constructive suggestions. The following responses might help address your concerns and questions about this paper:
> 1. **Weakness 1.** How does the backward process work in Algorithm 1? The sparse representation $\phi$ in Eq. 3 is not differentiable.
>
>      **Answer.** We sincerely thank you for perusing this work. We now further clarify how the backward process works in Algorithm 1 in the **revised version** and express how to address the concern about the differentiability of the sparse representation $\phi$.
>
>     In Algorithm 1, the backward process refers to the optimization of the dictionary and the transformed vector/matrix by minimizing the total final loss. This process also involves updating the sparse representation $\phi$ while keeping the dictionary $D$ and the input visual features fixed at the value obtained in the previous step.
>      Note that, we employ Elastic Net regularization to enforce sparsity in the dictionary learning/sparse coding problem $||x - D\phi|| + \lambda ||\phi||_0$.
>      &emsp;
>
>      Elastic Net regularization transforms the problem from NP-hard to a convex optimization problem with a closed-form solution for $\phi$.
>      The key objective function for dictionary learning, which decomposes data into a dictionary and its corresponding sparse representation, can be expressed as follows:
>
>      $\min_{D, \phi}J_{Dictionary\ Learning},$ and $J_{Dictionary\ Learning} = ||x - D\phi||_2^2 + \left( \beta ||\phi||_1 + \frac{\lambda}{2} ||\phi||_2^2 \right),$
>
>      where $x\in R^{n}$ represents the input visual feature, $D\in R^{n\times k}$ represents the dictionary, $\phi \in R^{k}$ represents the sparse code, and $\lambda_1$ and $\lambda_2$ are hyper-parameters used to balance $l$-1 and $l$-2 regularization.
>
>      Hence, given an initial unreliable dictionary
>      $\mathbf{D} \in \mathcal{S}(n,k): =$ { $\mathbf{D}\in \mathbb{R}_{*}^{n\times k}:diag(\mathbf{D}^T\mathbf{D})=\mathbf{I}_k$ }.
>
>      Then, by fixing dictionary D, the objective for obtaining $\phi$ becomes a regularized least square problem. Specifically, let the derivative of the dictionary learning objective function with respect to $\phi$ is zero, i.e.,  $\frac{\partial J}{\partial \phi} = 0$. We can derive an analytical expression for the sparse coding vector $\phi$, as shown in equation 3 of Algorithm 1 in the main document:
>      $\phi^*_i = \left(\mathbf{D}^{T} \mathbf{D} + \lambda \mathbf{I} \right)^{-1} \left(\mathbf{D}^{T} x_i + \beta \mathbf{v} \right)$.
>
>      In summary, the sparse code $\phi$ is fixed to the value obtained in the previous step. The backward propagation updating strategy then optimizes the dictionary, the diagonal transformed matrix $\mathbf{W_D}$, and transformed matrix $W_{\Phi}$ by minimizing the total final loss.  Subsequently, the dictionary $D$ and the input visual features are updated, and the corresponding sparse representations will be recomputed again.
>
> &emsp;
>
> 2. **Weakness 2.** "What are the mIoU values of the proposed method and the counterpart using self-attention on the ADE20K dataset for semantic segmentation?"
>
>      **Answer.**       We now place more experimental results in the rebuttal box at the top. More experiments are still on evaluated, and the results will be published as soon as possible.
>
>      &emsp;
>
> 3. **Weakness 3.** Since the paper proposes to learn sparse representation using dictionary learning-based visual attention, some example sparse representations should be shown in order to verify the effectiveness of the proposed method. While Fig. 3 visualized some attention maps for certain categories, the sparsity of the representation is still unknown.
>
>     **Answer.** We appreciate your careful comments. While Fig. 3 visualizes attention maps for certain categories, we acknowledge that the sparsity of the sparse representation $\Phi$ is not explicitly demonstrated in that particular figure.
>
>     In the **revised version**, we further include the results of hyperparameters $\lambda_2$ on the impact of the sparsity and accuracy of sparse representation. They show that the sparsity increases as $\lambda_2$ increases. According to [1, 2, 3] and existing results, it can be shown that the sparsity of the sparse representation enhances the discriminative performance of the dictionary as a feature base, while the task-driven transformation matrix selects well-disentangled features. Finally, this enables the model to capture more accurate visual attention and achieve better performance.
>
>     It is important to note that the primary focus of our paper is on introducing the concept of the dictionary learning-based visual attention module and showcasing its potential for capturing visual attention. Fig. 3 illustrates the ability of the proposed Dic-Attn module to capture relevant features for specific categories.
>
>     &emsp;
>
> 	[1] Fuchs, J-J. "On sparse representations in arbitrary redundant bases." _IEEE transactions on Information theory_ 50.6 (2004): 1341-1344.
>
> 	[2] Elad, Michael, and Alfred M. Bruckstein. "On sparse signal representations." _Proceedings 2001 international conference on image processing (Cat. No. 01CH37205)_. Vol. 1. IEEE, 2001.
>
> 	[3] Wei, Xian, Hao Shen, and Martin Kleinsteuber. "Trace quotient with sparsity priors for learning low dimensional image representations." _IEEE transactions on pattern analysis and machine intelligence_ 42.12 (2019): 3119-3135.

---

### Official Review · Reviewer_2tL6 · 2023-07-07

**Soundness:** 3 good
**Presentation:** 3 good
**Contribution:** 3 good
**Rating:** 7
**Confidence:** 5

**Summary:**

The paper proposes a novel attention mechanism called Dic-Attn that enhances the performance of deep vision models on various computer vision tasks. Specifically, the Dic-Attn module allows for the disentanglement of underlying nonlinear structural information in visual data, providing an intuitive and elegant way to exploit discriminative information and provide visual attention. Moreover, the proposed method uses shallow-depth attention modules, which are computationally efficient and can be easily integrated into existing deep learning architectures. The paper presents experimental results on several computer vision tasks, demonstrating the effectiveness of the proposed method and its superiority over several state-of-the-art methods.

**Strengths:**

- The proposed method is intuitive and novel. It combines the advantages of dictionary learning with deep neural networks. Compared to previous related methods like EMANet and HamNet, it cleverly avoids the gradient problem in the iterative procedure. From the methodology perspective, this method is beyond the acceptance bar.

- The proposed Dic-Attn module achieves promising performance on various computer vision tasks, including image classification and point cloud classification. Besides, the proposed method shows its efficiency over several attention methods on these tasks. For example, the paper compares the proposed method with several existing attention mechanisms, such as SE-Net, CBAM, and ECA-Net, and shows that the proposed method achieves better performance in terms of accuracy and efficiency.

- The paper provides a detailed analysis of the proposed method, including ablation studies and visualization of attention maps, to better understand its behavior and performance. For example, the paper shows that the proposed method can capture both local and global features of the input data and that the learned attention maps are interpretable and meaningful.

- The paper presents experimental results on several computer vision tasks, demonstrating the effectiveness of the proposed method and its potential for real-world applications. For example, the paper shows that the proposed method can be used for object recognition, scene understanding, and 3D point cloud classification, which are important tasks in computer vision.

**Weaknesses:**

# Inadequate referencing
The current manuscript overlooks referencing certain pertinent studies. The proposed method, while distinct in its own right, exhibits conceptual parallels with several prior works that should be acknowledged. Comparison to these similar works, which are neither referenced nor discussed, would greatly augment the overall discourse.
References:
1. Zhu Z, et al. Asymmetric non-local neural networks for semantic segmentation[C]//Proceedings of the IEEE/CVF international conference on computer vision. 2019: 593-602.
2. Li X, Zhong Z, Wu J, et al. Expectation-maximization attention networks for semantic segmentation[C]//Proceedings of the IEEE/CVF International Conference on Computer Vision. 2019: 9167-9176.

# Limited results presentation
I found the experimental results to be somewhat restrictive, with only one large-scale benchmark being demonstrated. For a more comprehensive understanding of the study's impact, I recommend showcasing the complete ADE2K results within the supplementary materials, instead of merely citing a single score within the body of the text.

# Visualization and explanation disparity
While Figures 3 in the main document and 1 in the supplementary material do an effective job of portraying the inner workings of the proposed method, the related explanation (lines 336-338) lacks clarity. Determining which attention map is closer to human visual attention or identifying the "referred objects" proves challenging due to the blandness of the given explanation. This area requires more in-depth exploration and improved elucidation to make the understanding of the model's performance more intuitive.

**Questions:**

For W_D and W_scale, why choose them as vectors instead of matrices?
Are there any ablation study to verify the advantages of this choice?

Besides, if they are vectors, they should be in bolded lower case.

**Limitations:**

No potential negative societal impact

---

> ### Author Rebuttal · Authors · 2023-08-10
>
> We appreciate your careful review, positive feedback and constructive suggestions. The following responses might help address your questions about this paper:
> 1. **Weakness 1.** "**Inadequate referencing...**
>
>    **Answer.** Thank you for your feedback. We have indeed reviewed the two papers you mentioned. To address this concern, we revised the manuscript to include a discussion and made sure to properly cite these related papers in the paper, so that readers can easily access and review the relevant literature.
>
> 	[1] takes advantage of the non-local network is potent to capture the long-range dependencies that are crucial for semantic segmentation, and relieves the shortcoming of non-local block. [1] reduces the middle dimension of the key and value in the attention without affecting the final output dimension, $R^{N\times N} \times R^{N\times C}\rightarrow R^{N\times S} \times R^{S\times C} \rightarrow R^{N\times C}$. In order to improve performance, [1] also adopts the Spatial Pyramid Pooling method to make multi-scale fusion with different $C$. It inspires us that multi-scale dictionaries may have potential for future research.
>
> 	[2] proposed a new attention mechanism for semantic segmentation, called Expectation Maximization Attention (EMA). The expectation (E) step works as estimating the expectation of the attention map and maximizing (M) step functions as updating the bases by maximizing the complete data likelihood. The output can be computed as the weighted sum of bases, where the weights are the normalized final attention maps.
>
> 	Our method is applying the dictionary learning method to obtain a set of basis vectors that can be used to represent data in a more efficient and compact way. The training of the task-driven dictionary atom transformed/weighting matrix $W_D$ can also be considered as a process of expectation maximization.
>     We appreciate your concern about the lack of references to certain pertinent studies in the current manuscript and your suggestion to include a comparison with these similar works.
>
>      &emsp;
>
>     [1] Zhu Z, et al. Asymmetric non-local neural networks for semantic segmentation[C]//Proceedings of the IEEE/CVF international conference on computer vision. 2019: 593-602.
>
>       [2] Li X, Zhong Z, Wu J, et al. Expectation-maximization attention networks for semantic segmentation[C]//Proceedings of the IEEE/CVF International Conference on Computer Vision. 2019: 9167-9176."
>
>      &emsp;
>
> 2. **Weakness 2.** "**Limited results presentation...**
>
>       We now place more experimental results in the rebuttal box at the top. More experiments are still on evaluated, and the results will be published as soon as possible and added in **the revised version**.
>
>      &emsp;
>
> 3. **Weakness 3.** "**Visualization and explanation disparity...**"
>
>     **Answer.** Thank you for this valuable suggestion. We agree that visual interpretations can help readers better understand the experiments' results. In the **updated version**, we do the following two revisions.
>
>     (1) We further clarify that the output Attention Maps are obtained through the reconstruction of the re-weighted dictionary and transformed sparse representations.
>
>     The diagonal transformed matrix $\mathbf{W_D}$ and transformed matrix $\mathbf{W_{\phi}}$  are specifically designed to re-weight the dictionary and transform its corresponding sparse representations, respectively. $\mathbf{W_D}$ operates in a vector-wise manner, where each element in the diagonal serves as a weight for the dictionary columns, i.e., atoms. On the other hand, $\mathbf{W_{\phi}}$ implements the sparse encoding of individual elements. These two parameters are updated via the backpropagation strategy and are thus driven by the task's final objective function. The dictionary $\mathbf{D}$ belongs to $\mathcal{S}(n,k): =$ { $\mathbf{D}\in \mathbb{R}_{*}^{n\times k}:diag(\mathbf{D}^T\mathbf{D})=\mathbf{I}_k$ } and the corresponding sparse representations is subject to the Elastic Net constraint, they are also data-driven.
>
>     (2) The output Attention Maps demonstrate that our proposed attention module exhibits sharper and more precise object boundaries, enhancing the segmentation accuracy of the model. Based on these observations, we infer that the advantage of our proposed attention module stems from the pipeline our proposed attention module introduces. It is suggested that the potential non-linear structural information disentanglement ability of dictionary learning can improve performance during segmentation.
>
>      &emsp;
>
> 4. **Question.** "For W_D and W_scale, why choose..."
>
>     **Answer.** We sincerely appreciate your careful review and constructive suggestions sincerely this paper. The typo has been corrected in the **revised version**.
>     The diagonal transformed matrix $\mathbf{W_D}$ and matrix $\mathbf{W_{\phi}}$  are specifically designed to re-weight the dictionary and transform its corresponding sparse representations, respectively.
>
>     (1) $\mathbf{W_D}$ operates in a vector-wise manner, where each element in the diagonal serves as a weight for the dictionary columns, i.e., atoms.
>
>     (2) On the other hand, $\mathbf{W_{\phi}}$ implements the sparse encoding of individual elements.
>     These transformation parameters are driven by the task objective function and are updated through the backpropagation process.

---

### Author Rebuttal · Authors · 2023-08-10

# More Experimental Results

## **Answer** to Related Questions including:

- Weakness 2 by Reviewer 2tL6
- Weakness 2 by Reviewer Vec1
- Weakness 2 by Reviewer nmhX
- Weakness 1 by Reviewer 1dqN
- Weakness 2 and Question 1 by Reviewer q2Dg

We appreciate your comments and your expectation for more explainable results or improved performance compared to the baseline. he following responses might help address your concerns about this paper:

**(1) Image Classification:** We have conducted additional experiments on the ImageNet-1k dataset for image classification. We use "Swin-T(WSA)" and "Swin-T(Dic-Attn)" to denote the Swin Transformer-Tiny model and Swin Transformer-Tiny model using the Windows Self-Attention module the proposed dictionary learning-based visual attention(Dic-Attn) module, respectively. Specifically, the added results of the classification task on ImageNet-1K are shown as follows:

#### **Table 1.  Additional classification results on ImageNet-1K.** Models are trained from scratch on ImageNet-1K. The input images are all resized to $224^2$. Our proposed Dic-Attn module brings up 1.37% gains in Top-1 accuracy compared with backbone Swin-T(WSA) at the beginning of training (50 epochs).

|Methods|Dataset|Resolution|Acc (50 Epochs)|Acc|
|---|:-:|:-:|:-:|:-:|
|Swin-T(WSA)|ImageNet-1K|$224\times 224$|70.29|81.23|
| Swin-T(Dic-Attn) |ImageNet-1K |$224\times 224$|71.66|81.29|

**(2) Robustness:** Regarding your suggestion to explore the application of sparse dictionary learning algorithms in robustness or denoising tasks, we agree that such experiments could provide valuable insights into the capabilities of our proposed method. Actually, we have reported the model performance on adversarial robustness under the Fast Gradient Sign Method (FGSM) and Projected Gradient Descent (PGD) in Section 4.2.1 in the main document. We set the FGSM attack with the perturbation magnitude $\epsilon = \frac{4}{255}$ and set $\epsilon= \frac{8}{255}$ or PGD, with iteration numbers $t = 10$ and step size $\alpha = \frac{2}{255}$. We classify images corrupted by adversarial attack FGSM and PGD, respectively. Then we evaluate the model performance by their attack classification accuracy.

#### **Table 2. Robustness Evaluation Results on CIFAR-100 dataset.** Comparison of our approach with backbones and other attention baseline methods, against various adversarial attacks.

| Model          | Attacks |      |       | # Param(M) |
| -------------- |:-------:|:----:|:-----:|:----------:|
| (From Scrach)  |  FGSM   | PGD  | Clean |            |
| RVT (SA)       |  0.20   | 0.10 | 64.80 |    8.29    |
| RVT (VARS-D)   |  10.58  | 3.80 | 62.20 |    7.68    |
| RVT (Dic-Attn) |  10.00  | 4.30 | 55.93 |    7.48    |

**(3) Denoising:** In response to your feedback, we have conducted experiments on the SIDD dataset for image denoising. The results of these experiments show that the transformer augmented with our proposed module outperforms the baseline in denoising performance.

#### **Table 3. Additional classification results on SIDD.**

| Methods             | PSNR | SSIM |
| ------------------- |:----:|:----:|
| Restormer(SA)       | 39.53 | 0.960  |
| Restormer(Dic-Attn) |  39.54 |0.958|

**(4)Inference Time:** Specifically, we add the Inference Time test to compare the efficiency of various attention modules explicitly. The added results of inference time are shown as follows:

#### **Table 4. Comparison of the inference time cost (ms) of various attention modules.**

  | Methods/Indicators | GPU Inference Time (ms) |
  | ------------------ |:-----------------------:|
  | SA                 |          82.20          |
  | DA                 |          64.40          |
  | $A^2$              |          8.00           |
  | HAM                |          7.70           |
  | ACF                |          22.60          |
  | Dic-Attn(Ours)     |          24.70          |

The added results report the inference time of our proposed module and the baseline models. We have added to Table 4 in the revised version, highlighting the advantages and disadvantages of our proposed model in terms of computational efficiency.

We appreciate valuable feedback, as it helps us improve the comprehensiveness and applicability of our experimental evaluation. More experiments are still on evaluated, and the results will be published as soon as possible and added to the **revised Manuscript and updated Supplementary Material**.

---

### Decision · Program_Chairs · 2023-09-21

**Decision:**

Accept (poster)

**Comment:**

The paper describes a novel attention module based on dictionary learning, which can be employed for several architectures in different computer vision tasks. The methods is demonstrated on two applications, e.g. image classification and point cloud classification. The reviewers expressed concerns in terms of writing and requested more experiments on other benchmarks (ImageNet). There was a constructive discussion between authors and reviewers on missing elements in the paper. All the reviewers appreciated the rebuttal, increasing their score. At the end of the process, there was an agreement among reviewers that the paper can be accepted.